# Generalized Animal Imitator:
# Agile Locomotion with Versatile Motion Prior

**Abstract:** The agility of animals, particularly in complex activities such as running, turning, jumping, and backflipping, stands as an exemplar for robotic system design. Transferring this suite of behaviors to legged robotic systems introduces essential inquiries: How can a robot be trained to learn multiple locomotion behaviors simultaneously? How can the robot execute these tasks with a smooth transition? And what strategies allow for the integrated application of these skills? This paper introduces the Versatile Instructable Motion prior (*VIM*) – a Reinforcement Learning framework designed to incorporate a range of agile locomotion tasks suitable for advanced robotic applications. Our framework enables legged robots to learn diverse agile low-level skills by imitating animal motions and manually designed motions with *Functionality* reward and *Stylization* reward. While the *Functionality* reward guides the robot's ability to adopt varied skills, the *Stylization* reward ensures performance alignment with reference motions. Our evaluations of the VIM framework span both simulation environments and real-world deployment. To our understanding, this is the first work that allows a robot to concurrently learn diverse agile locomotion tasks using a singular controller.

**Keywords:** Legged Robots, Imitation Learning, Learning from Demonstration

## 1 Introduction

Research efforts have been invested for years in equipping legged robots with agility comparable to that of natural quadrupeds. Picture a golden retriever gracefully maneuvering in a park: darting, leaping over obstacles, and pursuing a thrown ball. These tasks, effortlessly performed by many animals, remain challenging for contemporary legged robots. To accomplish such tasks, robots need not only master individual agile locomotion skills like running and jumping but also the capacity to adaptively select and configure these skills based on sensory inputs. We regard this kind of complicated task requiring highly agile locomotion skills as advanced parkour for legged robots. The inherent ability of quadrupeds to smoothly execute diverse locomotion skills across varied tasks inspires our pursuit of a control system with a general locomotion motion prior that includes these skills. In this direction, we introduce a novel RL framework, Versatile Instructable Motion prior (*VIM*) aiming to endow legged robots with a spectrum of reusable agile locomotion skills by integrating existing agile locomotion knowledge.

Historically, agile gaits[1, 2, 3] for legged robots have been sculpted using model-based or optimization methods. While promising, these methods demand significant engineering input and precise state estimation. Learning-based controllers enable robots to walk or run while addressing these limitations, although they still fall short of agility. Imitation-based controllers are also proposed to learn from motion sequences from animals [4] or optimization methods [5]. Research on incorporating sensory information, such as visual observations [6, 7, 8, 9, 10, 11, 12] or elevation maps [13, 14] further enables legged robots to traverse complex terrain like stones. In spite of the encouraging results, most of these works focus on building a single controller from scratch, even though much of the learned

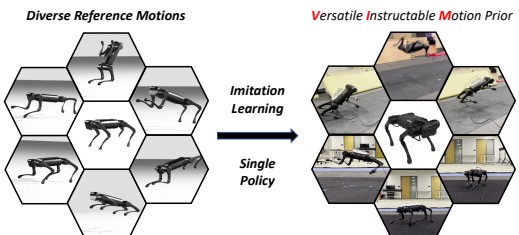

Figure 1: **Learning Agile Locomotion Skills from Reference Motions:** Our system learns a single instructable motion prior from a diverse reference motion dataset.

locomotion skills could be shared across tasks. Recent works start building a reconfigurable low-level motion prior [15, 16, 17, 18, 19, 20] for downstream applications. However, the previous methods failed to make the best use of existing skills to learn diverse locomotion skills with high agility.

In this work, we focus on building low-level motion prior to utilize existing locomotion skills in nature and previous optimization methods, and learn multiple highly agile locomotion skills simultaneously, as shown in Figure 1. Even though we cannot fully comprehend the agility of animals and lack a unified framework for model-based controls, we recognize that motion sequences offer a consistent representation of diverse agile locomotion skills. Our motion prior extracts and assimilates a range of locomotion skills from reference motions, effectively mirroring their dynamics. These references comprise motion capture (mocap) sequences from quadrupeds, augmented generative model sequences complementing mocap data, and optimized motion trajectories. Throughout the training phase, we translate varied reference motion clips into a unified latent command space, guiding the motion prior to recreate locomotion dynamics based on these latent commands and the robot's inherent state.

For legged robots, we define a locomotion skill as the ability of the robot to produce a specific trajectory. To break down the intricacies of movement, we classify it into two primary facets: *Functionality* and *Style*. *Functionality* pertains to the fundamental movement objectives a robot aims to achieve, such as advancing forward at a predefined speed. *Style*, in contrast, delves into the specific mechanics of how a robot accomplishes a task. To illustrate, two robots might be programmed to progress at an identical speed, but the intricacies of their movement—like step size or frequency—might differ considerably. Simultaneously instructing a robot in both these domains is nontrivial[21]. Drawing inspiration from how humans learn complicated tasks, especially in fields demanding physical prowess like athletics, we identify three core feedback modalities: objective performance metrics, qualitative assessments, and granular kinematic guidance. Adopting this structured feedback approach, our robot starts with mastering the basic functional objective and subsequently turns into refining the detailed locomotion gaits.

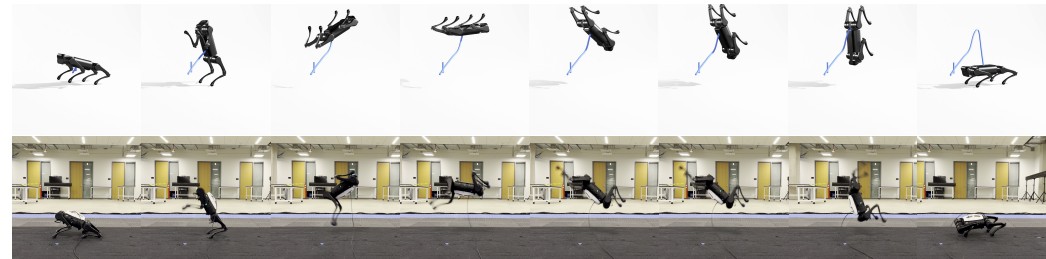

Figure 2: **Real-Robot Trajectory.** Our robot exhibits back-flipping skill in the second row by imitating the reference motion in the first row.

By incorporating diverse reference motions and our reward design, our Versatile Instructable Motion prior (*VIM*) learns diverse agile locomotion skills and makes them available for intricate downstream tasks. With our VIM, we enable legged robots to perform Advanced Robotics Parkour in the real world. We also evaluate our method in the simulation and real world, as Figure 2. Our method significantly outperforms baselines in terms of final performance and sample efficiency.

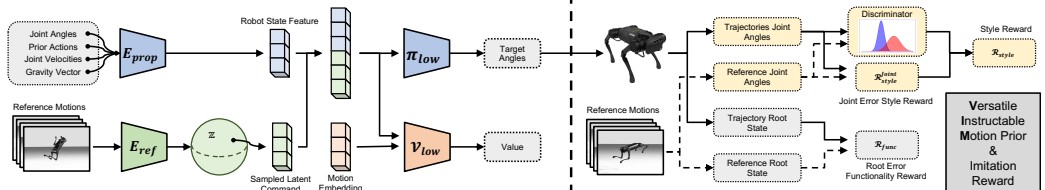

Figure 3: **V**ersatile **I**nstructable **M**otion prior (*VIM*): Reference motion encoder maps reference motions into latent skill space indicating target robot pose and low-level policy output motor command. **Reward Design:** Our includes *Functionality* reward and *Style* reward.

## 2    Related Work

**Blind Legged Locomotion:** Classical legged locomotion controllers [22, 23, 24, 25, 1, 26] based on model-based methods [27, 28, 29, 30, 31, 32, 33] and trajectory optimization [34, 3] have shown promising results in diverse tasks with high levels of agility. Nonetheless, these methods normally come with considerable engineering tuning for the specific task, high computation requirements during deployment, or fragility to complex dynamics. Learning-based methods [6, 35, 13, 36, 37, 38] controllers are proposed to offer robust and lightweight controllers for deployment at the cost of offline computation. Peng et al. [39] developed a controller producing non-agile life-like gaits by imitating animal. Though previous works offer robust or agile locomotion controllers across complex environments, these works focus on finishing a single task at a time without reusing previous experience. Smith et al. [40] utilize existing locomotion skills to solve specific downstream tasks. Vollenweider et al. [41] utilize multiple AMP [42] to develop a controller to solve a fixed task set. In this paper, our motion prior captures diverse agile locomotion skills from reference motions generated by trajectory optimization and provides them for intricate future downstream tasks.

**Motion Priors:** Due to the notorious low sample efficiency and considerable effort required for reward engineering of RL, low-level skill pretraining has drawn growing attention in recent years. Singh et. al [15] utilize the flow-based model to build an actionable motion prior with motion sequences generated by scripts. More recent works [16, 17, 18, 19, 43, 20] focus on building low-level motion prior for downstream tasks but fail to include diverse highly agile locomotion skills. In this work, we build motion prior with reference motions consisting of mocap sequences, synthesized motion sequences, and trajectories from optimization methods and learn multiple highly agile locomotion skills with a single controller.

## 3    Learn Versatile Instructable Motion Prior

We present the **V**ersatile **I**nstructable **M**otion prior (*VIM*), depicted in Figure 3, designed to acquire a wide range of agile locomotion skills concurrently from multiple reference motions. The development of our motion prior involves three stages: assembling a comprehensive dataset of reference motions sourced from diverse origins, crafting a motion prior that processes varying reference motions and the robot's proprioceptive feedback to generate motor commands, and finally, utilizing an imitation-based reward mechanism to effectively train this motion prior.

### 3.1    Reference motion dataset

Our primary objective was to curate a skill set for the robot that covers diverse functions and agility levels, equipping it to handle complex downstream tasks. Our dataset encompasses reference motions for locomotion skills, including but not limited to canter, pace, walk, trot, turns, backflips, and various jumps. These reference motions are derived from:*(a)* mocap data of quadrupeds, specifically a subset from previous work [44], despite its inherent challenges like noise due to the unpredictability of animal behavior; *(b)* synthesized (Syn) motions generated using a generative model [44], aimed at enhancing dataset diversity by capturing challenging locomotion actions;*(c)* motions crafted through trajectory optimization methods (Opt).

110 To address the methodology disparities between quadrupeds and our robot, we retarget both mocap
111 and synthesized sequences to our robot as Peng et al. [4]. While mocap and synthesized motions
112 offer extensive data, not all sequences may be practically achievable by the robot. Thus, our dataset
113 is supplemented with motion sequences from trajectory optimization, emphasizing intricate moves
114 like jumps and backflips. The comprehensive reference motion list can be found in Table 2. Each
115 trajectory in our dataset, represented as $(s_0^{\text{ref}}, s_1^{\text{ref}}, \cdots, s_T^{\text{ref}})$, focuses on the robot's trunk and joint
116 movements, excluding specific motor commands which are absent in the captured and synthesized
117 data. We denote the dataset as $\mathcal{D} = \{(s_0^{\text{ref}}, s_1^{\text{ref}}, \cdots, s_T^{\text{ref}})_i\}_{i=1}^N$.

## 3.2 Motion Prior Structure

119 Our motion prior consists of a reference motion encoder, and a low-level policy. Reference motion
120 encoder maps varying reference motions into a condensed latent skill space, and low-level policy
121 utilizes our imitation reward, reproduces the robot motion given a latent command.

122 **Reference motion encoder:** Our reference motion encoder $\mathbb{E}_{\text{ref}}(\cdot)$ maps segments of reference motion
123 to latent commands in a latent skill space that outline the robot's prospective movement. These
124 segments span both imminent and distant future states, expressed as $\hat{s}_t^{\text{ref}} = \{s_{t+1}^{\text{ref}}, s_{t+2}^{\text{ref}}, s_{t+10}^{\text{ref}}, s_{t+30}^{\text{ref}}\}$.
125 We model the latent command as a Gaussian distribution $\mathcal{N}(\mathbb{E}_{\text{ref}}^\mu(\hat{s}_t^{\text{ref}}), \mathbb{E}_{\text{ref}}^\sigma(\hat{s}_t^{\text{ref}}))$ from which we
126 draw a sample at each interval to guide the low-level policy.

127 To maintain a *temporal-consistent* latent skill space, our training integrates an information bottle-
128 neck [45, 46] objective $L_{\text{AR}}$, where the prior follows an auto-regressive model [47]. Specifically,
129 given the sampled latent command for the previous time step $z_{t-1}$, we minimize the KL divergence
130 between the current latent Gaussian distribution and a Gaussian prior parameterized by $z_{t-1}$,

$$L_{\text{AR}}(\hat{s}_t^{\text{ref}}, z_{t-1}) = \beta \text{KL} \left( \mathcal{N}(\mu_t, \sigma_t^2) \| \mathcal{N}(\alpha z_{t-1}, (1 - \alpha^2)I) \right),$$

131 where $\alpha = 0.95$ is the scalar controlling the effect of correlation, $\beta$ is the coefficient balancing
132 regularization.

133 **Low-level policy training:** Our low-level policy $\pi_{\text{low}}$ takes latent command $z_t$ representing the
134 desired robot pose and robot's current proprioceptive state $s_t$ as input, and outputs actual motor
135 commands $a_t$ for the robot, where $s_t$ is encoded with a proprioception encoder $\mathbb{E}_{\text{prop}}$. We train
136 low-level policy and reference motion encoder using PPO [48] in an end-to-end manner. Additionally,
137 we introduce a motion embedding for the critic to distinguish diverse reference motions. Episodes
138 initiate with randomized starting time steps from the dataset to avert overfitting and conclude when
139 the root pose tracking error escalates beyond an acceptable range.

## 3.3 Imitation Reward for Functionality and Style

141 Given the formulation of our motion prior, the robot learns diverse agile locomotion skills with our
142 imitation reward and reward scheduling mechanics. Our reward offers consistent guidance, ensuring
143 the robot captures both the functionality and style inherent to the reference motion.

144 **Learning Skill Functionality:** To mirror the functionality of the reference motion, we translate the
145 root pose discrepancy between agent trajectories and reference motion into a reward. The functionality
146 reward $r_{\text{func}}$ is subdivided into tracking rewards for robot root position $r_{\text{func}}^{\text{pos}}$ and orientation $r_{\text{func}}^{\text{ori}}$.
147 Recognizing the distinct importance of vertical movement in agile tasks, the root position tracking is
148 further split into rewards for vertical $r_{\text{func}}^{\text{pos-z}}$ and horizontal movements $r_{\text{func}}^{\text{pos-xy}}$.

$$r_{\text{func}}(s_t, \hat{s}_t^{\text{ref}}) = w_{\text{func}}^{\text{ori}} * r_{\text{func}}^{\text{ori}} + w_{\text{func}}^{\text{pos-xy}} * r_{\text{func}}^{\text{pos-xy}} + w_{\text{func}}^{\text{pos-z}} * r_{\text{func}}^{\text{pos-z}}$$

149 The specific formulation of our functionality rewards is provided as follows, which is similar to
150 previous work[4].

$$r_{\text{func}}^{\text{ori}}(s_t, \hat{s}_t^{\text{ref}}) = \exp\left(-10 \left\| \hat{\mathbf{q}}_t^{\text{root}} - \mathbf{q}_t^{\text{root}} \right\|^2\right)$$

151

$$r_{\text{func}}^{\text{pos-xy}}(s_t, \hat{s}_t^{\text{ref}}) = \exp\left(-20 \left\| \hat{\mathbf{x}}_t^{\text{root-xy}} - \mathbf{x}_t^{\text{root-xy}} \right\|^2\right) r_{\text{func}}^{\text{pos-z}}(s_t, \hat{s}_t^{\text{ref}}) = \exp\left(-80 \left\| \hat{\mathbf{x}}_t^{\text{root-z}} - \mathbf{x}_t^{\text{root-z}} \right\|^2\right)$$

where $\mathbf{q}$, $\hat{\mathbf{q}}$ and $\mathbf{x}$, $\hat{\mathbf{x}}$ denote the root orientation and position from both the robot and reference motion, respectively. Notably, in contrast to previous work [4], we allocate a greater emphasis on root height in our reward, crucial for mastering agile locomotion skills such as backflips and jumps.

**Learning Skill Style:** Capturing the style of a reference motion, in addition to its functionality, enriches the application by meeting criteria such as energy efficiency, robot safety, and facilitating human-robot interaction. Drawing inspiration from how humans learn - starting by emulating the broader style before focusing on intricate joint movements - our robot first mimics the broader locomotion style with an adversarial style reward and later refines its technique with a joint angle tracking reward.

**Adversarial Stylization Reward:** To swiftly encapsulate the style of the locomotion skill, we train distinct discriminators $D_i$, $i = 1..n$ for all $n$ reference motions separately to distinguish robot transitions from the transition of the specific reference motion[42, 41] and use the output to provide high-level feedback to the agent. Specifically, our discriminator is trained with the following objective:

$$\underset{D_i}{\text{argmin}} \; \underset{d_i^{\mathcal{M}}(s,s')}{\mathbb{E}} \left(D_i(s,s') - 1\right)^2 + \underset{d_i^{\pi}(s,s')}{\mathbb{E}} \left(D_i(s,s') + 1\right)^2$$

where $d_i^{\mathcal{M}}(s,s')$, $d_i^{\pi}(s,s')$ denote the transition distribution of the dataset and policy for $i$th reference motion respectively.

For each reference motion, the likelihood from the discriminator is then converted to a reward with:

$$r_{\text{style}}^{\text{adv}}(s_t, s_t') = 1 - \frac{1}{4} * \left(1 - D(s_t, s_t')\right)^2.$$

Initially, our adversarial stylization reward provides dense reward and enables the robot to learn a credible gait, but it can not provide more detailed instructions as the training proceeds, which leads to mode collapse and unstable training.

**Joint Angle tracking Reward:** On the other end, joint angle tracking reward [49, 17] provides sparse but stable instruction for the robot to mimic the gait of reference motion. Similar to our root pose tracking reward, our joint angle tracking reward has the following formulation:

$$r_{\text{style}}^{\text{joint}}(s_t, \hat{s}_t^{\text{ref}}) = \exp\left(-5 \sum_{j \in \text{joints}} \left\|\hat{\mathbf{q}}_t^j - \mathbf{q}_t^j\right\|^2\right) + \exp\left(-20 \sum_{f \in \text{feet}} \left\|\hat{\mathbf{x}}_t^f - \mathbf{x}_t^f\right\|^2\right)$$

where $\mathbf{q}_t^j$, $\hat{\mathbf{q}}_t^j$ are the joint angle of robot and reference motion and $\mathbf{e}_t^f$, $\hat{\mathbf{e}}_t^f$ are the end-effector positions of robot and reference motion.

When learning diverse agile locomotion skills, only combining the joint angle tracking reward and functionality reward leads to the failure of tracking functionality or tracking the style of reference motion. Since different locomotion skills are sensitive to different rewards.

**Stylization Reward Scheduling:** To take the best of both worlds, we propose to use both adversarial stylization reward and joint angle tracking reward with a balanced scheduling mechanism. Considering the discriminator as a "coach", we utilize the mean adversarial reward as an indication of how the coach is satisfied with the current performance. When it's not satisfied with the current performance of the robot, it provides detailed instruction for the robot to learn. Specifically our stylization reward follows:

$$r_{\text{style}}(s_t, \hat{s}_t^{\text{ref}}) = w_{\text{style}}^{\text{adv}} * r_{\text{style}}^{\text{adv}} + w_{\text{style}}^{\text{joint}} * r_{\text{style}}^{\text{joint}} + w_{\text{style}}^{\text{adv}} * (1 - \underset{s_t \in S}{\mathbb{E}} (r_{\text{style}}^{\text{adv}}(s,s'))) * r_{\text{style}}^{\text{joint}}$$

With the given formulation, our stylization reward provided dense rewards during the beginning of training, enabling the robot to quickly catch the essence of different agile locomotion skills. Our stylization reward also provides detailed instruction as the training proceeds, enabling the robot to refine its gait and lead to more stable training.

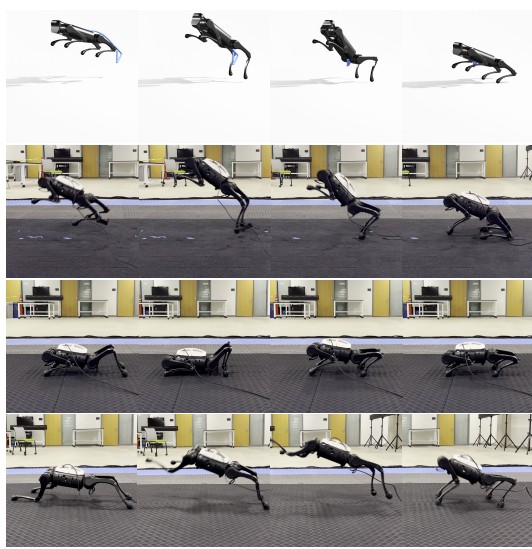

Figure 4: **Real World `Jump Forward` Trajectory Comparison:** Each row represents a single trajectory (From top to bottom: Reference Motion, VIM, GAIL, Motion Imitation).

Table 1: **Evaluation of Motion Prior in Simulation:** We compare Horizontal and Vertical Root Position (Root Pos (XY), Root Pos (Height)), Root Orientation (Root Ori), Joint Angle, and End Effector Position (EE Pos) tracking errors and RL objectives of all methods. Our methods outperform all baselines in terms of smaller tracking errors, higher episodic returns, and longer episode lengths. GAIL baseline shows a smaller root position tracking error since it can't follow the reference motion leading to early termination of the episode.

| Method | Tracking Error ↓ | | | | | RL Objectives ↑ | |
|---|---|---|---|---|---|---|---|
| | Root Pos (XY) | Root Pos (Height) | Root Ori | Joint Angle | EE Pos | Episode Return | Episode Length |
| VIM | $\mathbf{1.24}_{\pm\mathbf{0.62}}$ | $0.01_{\pm 0.02}$ | $0.11_{\pm 0.06}$ | $\mathbf{0.08}_{\pm\mathbf{0.06}}$ | $\mathbf{0.03}_{\pm\mathbf{0.03}}$ | $13.313_{\pm 11.48}$ | $166.783_{\pm 120.217}$ |
| VIM (w/o Scheduling) | $1.28_{\pm 0.67}$ | $0.009_{\pm 0.0123}$ | $\mathbf{0.1}_{\pm\mathbf{0.06}}$ | $0.1_{\pm 0.08}$ | $0.05_{\pm 0.04}$ | $\mathbf{13.963}_{\pm\mathbf{11.395}}$ | $\mathbf{179.047}_{\pm\mathbf{121.788}}$ |
| Motion Imitation | $1.39_{\pm 0.66}$ | $\mathbf{0.0077}_{\pm\mathbf{0.0114}}$ | $0.11_{\pm 0.05}$ | $0.25_{\pm 0.14}$ | $0.14_{\pm 0.08}$ | $9.536_{\pm 9.049}$ | $143.393_{\pm 114.514}$ |
| GAIL | $1.04_{\pm 0.86}$ | $0.03_{\pm 0.03}$ | $0.13_{\pm 0.05}$ | $0.17_{\pm 0.1}$ | $0.09_{\pm 0.05}$ | $3.586_{\pm 6.166}$ | $54.723_{\pm 75.984}$ |

### 3.4 Solving Downstream Tasks with Motion Prior:

For intricate tasks like jumping over gaps, addressing them from scratch presents challenges including acquiring necessary agile locomotion skills, such as jumping and running within limited interactions, and the intensive engineering needed to harmonize the reward for top-tier tasks while regularizing the robot's motion. With a low-level motion prior, robots can instantly harness existing skills encapsulated within the prior and channel their efforts into high-level strategizing. For each distinct downstream task, we train a high-level policy $\pi_{\text{high}}$ takes the high-level observation $\mathbf{o}_{\text{high}}$ and outputs latent command for low-level motion prior to utilize the existing agile locomotion skills: $a_t = \pi_{\text{low}}\big(\pi_{\text{high}}(\mathbf{o}_{\text{high}}, \mathbf{E}_{\text{prop}}(s_t))\big)$.

## 4 Experiments

### 4.1 Evaluation of Learned Motion Priors

Our system's proficiency in learning a range of agile locomotion skills from the reference motion dataset (discussed in Sec 3.1) is initially assessed.

**Baselines**: We benchmark our method against two representative baselines: Motion Imitation [4, 17, 20] baseline represents a thread of recent works whose imitation rewards are defined solely with errors between current robot states and the corresponding reference states. Generative Adversarial Imitation Learning (GAIL) baseline represents a thread of recent work [18], whose imitation reward is solely provided by the discriminator trained to distinguish trajectories generated by the policy from the ground truth reference motions. Given that our reference motions consist only of state sequences, they offer less supervision compared to expert action sequences, rendering motion prior learning more challenging. Each method trains for $2 \times 10^9$ iterations across 3 random seeds. Both our technique

Table 2: **Evaluation of Motion Prior in Real (Left):** We collect representative metrics for different locomotion skills with corresponding metrics from reference motion. $N/A$ denotes completely failed skills in real. **Full Reference Motion List (Right)**

| Metrics | VIM | Motion Imitation | GAIL | Reference Motion | Skill Name | Source |
|---|---|---|---|---|---|---|
| Height (Jump While Running) $(m)$ | **0.50**±0.003 | 0.42±0.01 | 0.41±0.04 | 0.53±0.005 | Walk | Mocap |
| Height (Jump Forward) $(m)$ | **0.44**±0.01 | 0.42±0.01 | 0.27±0.006 | 0.59±0.006 | Trot | Mocap |
| Height (Jump Forward (Syn)) $(m)$ | **0.52**±0.01 | $N/A$ | $N/A$ | 0.55±0.007 | Jump while Running | Mocap |
| Height (Backflip) $(m)$ | **0.62**±0.01 | 0.49±0.01 | $N/A$ | 0.60±0.005 | Right Turn | Mocap |
| Distance (Jump While Running) $(m)$ | **0.48**±0.08 | 0.35±0.02 | 0.40±0.003 | 0.56±0.008 | Left Turn | Mocap |
| Distance (Jump Forward) $(m)$ | **0.76**±0.05 | 0.40±0.01 | 0.10±0.002 | 0.82±0.003 | Backflip | Opt |
| Distance (Jump Forward (Syn)) $(m)$ | **0.49**±0.04 | $N/A$ | $N/A$ | 0.54±0.007 | Jump Forward (Syn) | Syn |
| Linear Velocity (Pace) $(m/s)$ | **0.76**±0.01 | 0.97±0.07 | 0.50±0.02 | 0.72±0.05 | Left Turn (Syn) | Syn |
| Linear Velocity (Canter) $(m/s)$ | **1.49**±0.15 | $N/A$ | $N/A$ | 3.87±0.17 | Jump Forward | Opt |
| Linear Velocity (Walk) $(m/s)$ | 0.90±0.04 | **0.96**±0.06 | 0.53±0.58 | 0.97±0.42 | Canter | Mocap |
| Linear Velocity (Trot) $(m/s)$ | 1.33±0.17 | **1.05**±0.02 | 0.93±0.01 | 1.16±0.12 | Pace | Mocap |
| Angular Velocity (Left Turn) $(rad/s)$ | 1.71±0.04 | 0.00±0.00 | **0.91**±0.04 | 1.01±0.05 | | |
| Angular Velocity (Right Turn) $(rad/s)$ | 0.81±0.02 | **0.62**±0.02 | 0.63±0.05 | 0.41±0.09 | | |
| Joint Angle Tracking Error $(rad^2/joint)$ | **0.10**±0.08 | 0.27±0.16 | 0.22±0.10 | - | | |

and the Motion Imitation baseline adopt identical reward scales for all motion error-tracking rewards. Likewise, our approach and GAIL maintain the same scale for adversarial stylization rewards.

**Simulation Evaluation:** In the simulation, we measure average imitation tracking errors for various agile locomotion skills, episode returns, and trajectory lengths across random seeds. Specifically as listed in Table 1, where the tracking error of root pose represents the ability of the robot to reproduce the locomotion skill, and the tracking error of joint angle and end effector position represents the ability of the robot to mimic the style of reference motion. Our method achieves a similar root pose tracking error as the motion imitation baseline with a much smaller joint angle tracking error. This shows our method striking a balance between functionality and style, superior to the motion imitation baseline that focuses solely on functionality. Meanwhile, the GAIL baseline failed to learn the functionality of the reference motions which leads to short episode length and the least episode return. We surmise that the GAIL baseline's inadequacy arises for two main reasons: First, exclusive reliance on adversarial stylization reward does not offer temporally consistent guidance throughout skill learning due to misaligned rewards across timesteps. Second, the mode collapse issue inherent in adversarial training hinders the robot from mastering highly agile skills, such as backflipping. The shortcomings of the Motion Imitation baseline may stem from the challenges of balancing different terms and selecting suitable hyperparameters when concurrently learning multiple agile locomotion skills. Comparing our VIM with and without stylization reward scheduling, we find the former exhibits enhanced style tracking performance, underscoring the value of stylization reward scheduling in refining robot gait tracking.

**Real World Evaluation:** We gauge learned agile locomotion skills in real-world scenarios. Due to the lack of precise robot pose, we resort to specific metrics tailored to different locomotion skills, detailed in Table 2. For `Jump While Running` & `Jump Forward` & `Jump Forward (Syn)` & `Backflip`, we measure the jumping height and jumping distance. For `Pace` & `Canter` & `Walk` & `Trot` and `Left Turn` & `Right Turn`, we measure the linear and angular velocity, respectively. Results reveal that our method retains most of the reference motion functionality. The only significant deviation, observed in the `Canter` motion, arises from inherent differences between animal movement (its source) and our robot's capabilities. Even with comparable root pose tracking errors in simulations, our method outshines the Motion Imitation baseline in real-world metrics like jumping height, distance, and velocity tracking error. This suggests that mirroring the style of the reference motion improves sim2real transfer for natural gaits. GAIL baseline struggled to reproduce most real-world locomotion skills. A visual comparison of real-world trajectories is available in Figure 4, showing our method's superiority in capturing both motion functionality and style.

## 4.2 Evaluation on Downstream Tasks

**Downstream Tasks:** Our task suite comprises: `Following Command`: This involves directing the robot to move with specific linear and angular velocities, sampled uniformly between $0 \sim 2$ m/s and $-2 \sim 2$ rads/s. In our motion prior, the robot is trained to move and turn at the reference motion's speed; hence, to follow a command precisely, the high-level policy should smoothly interpolate between different speeds. `Jump Forward`: This task requires the robot to execute a jump during a forward run. We have adapted a subset of jumping rewards from CAJun [50] to evaluate policy interpolation between jumping and running motions within a fixed timeframe. `Following`

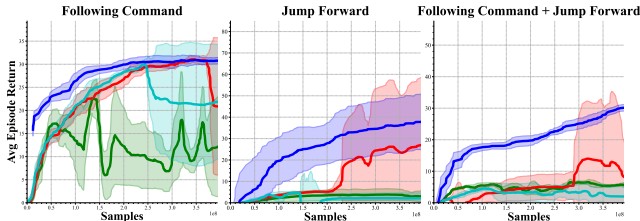

Figure 5: **Downstream Tasks Evaluation in Simulation:** Solid line and shaded area denote the mean and std across random seeds. Our system outperforms all baselines on all tasks.

Table 3: **Downstream Tasks Evaluation in Real:** We compare `Following Command + Jump Forward` policies of all methods in real, and $N/A$ denotes completely failed skills in real. Our methods outperform all baselines in real for most metrics.

| Metrics (Vel for Velocity) | Ours | AMP | PPO | HRL |
|---|---|---|---|---|
| Max Linear Vel $(m/s)$ | $\mathbf{1.78}_{\pm \mathbf{0.13}}$ | $1.74_{\pm 0.21}$ | $1.75_{\pm 0.26}$ | $1.70_{\pm 0.08}$ |
| Max Angular Vel (Left) $(rad/s)$ | $1.78_{\pm 0.004}$ | $1.07_{\pm 0.09}$ | $\mathbf{2.24}_{\pm \mathbf{0.05}}$ | $0.00_{\pm 0.00}$ |
| Max Angular Vel (Right) $(rad/s)$ | $\mathbf{2.05}_{\pm \mathbf{0.02}}$ | $0.83_{\pm 0.09}$ | $1.75_{\pm 0.19}$ | $0.95_{\pm 0.37}$ |
| Jump Distance $(m)$ | $\mathbf{0.50}_{\pm \mathbf{0.07}}$ | $0.00_{\pm 0.00}$ | $N/A$ | $N/A$ |
| Jump Height $(m)$ | $\mathbf{0.50}_{\pm \mathbf{0.02}}$ | $0.38_{\pm 0.01}$ | $N/A$ | $N/A$ |

`Command + Jump Forward`: Here, the robot must either jump forward or adjust to changing commanded speeds. To optimize episode return, the robot should not only use the agile locomotion skills from the reference motion dataset but also develop unobserved skills like executing sharp turns.

**Baselines:** Considering the baseline's subpar performance in low-level motion prior training, we compare our system with three representative baselines without pre-trained low-level controller: **PPO** [48]: Demonstrates controllers trained exclusively on downstream task rewards. **AMP** [42] utilize existing reference motion to provide styling reward in an adversarial imitation learning manner and learn the policy for the downstream task while mimicking the behavior of reference motions. Jain et al.**Hierarchical Reinforcement Learning (HRL)** adapts from [51] which learns a high-level policy sending latent commands to a low-level motor controller. HRL resembles a broad category of prior works that decompose temporally extended reasoning into sub-problems [52, 53, 54, 55, 56, 57, 58]. For a fair comparison, we made modifications like removing the trajectory generator in [51], using PPO for AMP and HRL, and supplying full reference motion data to AMP and HRL integrated with AMP.

**Evaluation in Simulation & Real World:** We train all methods on each downstream task for $4 \times 10^8$ environment samples with 3 random seeds. The simulation results are detailed in Figure 5, and real-world results are provided in Table 3. For the `Following Command` task, while all methods mastered basic locomotion, ours excelled in efficiency and smoothly transitioned between diverse linear and angular velocities. The other tasks, `Jump Forward` and `Following Command + Jump Forward`, demanded advanced jumping abilities, which baselines couldn't emerge. These baseline methods either continuously moved forward, remained grounded when prompted to jump, or toppled to evade energy consumption penalties. In contrast, our system seamlessly bridged jumping and running actions, securing the highest episode return. Despite providing with a comprehensive reference motion dataset, baselines couldn't harness the skills. This shortcoming possibly stems from the challenge of deriving agile locomotion skills from the dataset using only adversarial stylization rewards, mirroring the GAIL baseline's poor performance in low-level motion prior training.

# 5 Conclusion

In this paper, we propose Versatile Instructable Motion prior (*VIM*) which learns agile locomotion skills from diverse reference motions with a single motion prior. Our results in simulation and in the real world show that our VIM captures both the functionality and the style of locomotion skills from reference motions. Our VIM also provides a temporally consistent and compact latent skill space representing different locomotion skills for different downstream tasks. With agile locomotion skills in our VIM, complex downstream tasks can be solved efficiently with minimum human effort.

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
