# OpenReview forum: "Generalized Animal Imitator: Agile Locomotion with Versatile Motion Prior"
_robot-learning.org/CoRL/2023/Workshop/TGR — CoRL 2023 Workshop TGR Poster_

### Official Review · Reviewer_Hh56 · 2023-10-19

**Rating:** 7
**Confidence:** 3

**Review:**

This paper introduces the Versatile Instructable Motion prior (VIM),  a Reinforcement Learning framework designed to incorporate a range of agile locomotion tasks. While the work aims at learning diverse and versatile locomotion skills, it's not manifestly clear how such an approach can be directly connected to scalable robot learning. However, the focus on extensive locomotion skills (in contrast to most other submissions that focus on manipulation) makes it a good fit to be presented in the workshop.

---

### Official Review · Reviewer_WojE · 2023-10-20

**Rating:** 7
**Confidence:** 3

**Review:**

This paper learns versatile motion priors that can be applied to downstream tasks and has been demonstrated through real-world experiments. I think the versatile motion priors learning fit in our theme.

---

### Decision · Program_Chairs · 2023-10-20

**Decision:**

Accept (Poster)

**Comment:**

Great paper and closely aligned topic!